

# DiscoSnp-RAD: de novo detection of small variants for RAD-Seq population genomics

Jérémy Gauthier[1], Charlotte Mouden[1], Tomasz Suchan[2],
Nadir Alvarez[3,4], Nils Arrigo[3], Chloé Riou[1], Claire Lemaitre[1] and
Pierre Peterlongo[1]

[1] Univ. Rennes, Inria, CNRS, IRISA, Rennes, France
[2] W. Szafer Institute of Botany, Polish Academy of Sciences, Krakow, Poland
[3] Department of Ecology and Evolution, University of Lausanne, Lausanne, Switzerland
[4] Natural History Museum of Geneva, Geneva, Switzerland

## ABSTRACT

Restriction site Associated DNA Sequencing (RAD-Seq) is a technique characterized by the sequencing of specific loci along the genome that is widely employed in the field of evolutionary biology since it allows to exploit variants (mainly Single Nucleotide Polymorphism—SNPs) information from entire populations at a reduced cost. Common RAD dedicated tools, such as *STACKS* or *IPyRAD*, are based on all-vs-all read alignments, which require consequent time and computing resources. We present an original method, DiscoSnp-RAD, that avoids this pitfall since variants are detected by exploiting specific parts of the assembly graph built from the reads, hence preventing all-vs-all read alignments. We tested the implementation on simulated datasets of increasing size, up to 1,000 samples, and on real RAD-Seq data from 259 specimens of *Chiastocheta* flies, morphologically assigned to seven species. All individuals were successfully assigned to their species using both STRUCTURE and Maximum Likelihood phylogenetic reconstruction. Moreover, identified variants succeeded to reveal a within-species genetic structure linked to the geographic distribution. Furthermore, our results show that DiscoSnp-RAD is significantly faster than state-of-the-art tools. The overall results show that DiscoSnp-RAD is suitable to identify variants from RAD-Seq data, it does not require time-consuming parameterization steps and it stands out from other tools due to its completely different principle, making it substantially faster, in particular on large datasets.

# INTRODUCTION

Next-generation sequencing and the ability to obtain genomic sequences for hundreds to thousands of individuals of the same species has opened new horizons in population genomics research. This has been made possible by the development of cost-efficient approaches to obtain sufficient homologous genomic regions, by reproducible genome complexity reduction and multiplexing several samples within a single sequencing run (*Andrews et al., 2016*). Among such methods, the most widely used over the last decade is "Restriction-site Associated DNA sequencing" (RAD-Seq). It uses restriction enzymes to

Corresponding author
Pierre Peterlongo,
pierre.peterlongo@inria.fr

digest DNA at specific genomic sites whose adjacent regions are then sequenced. This approach encompasses various methods with different intermediate steps to optimize the genome sampling, for example, ddRAD (*Peterson et al., 2012*), GBS (*Elshire et al., 2011*), 2b-RAD (*Wang et al., 2012*), 3RAD/RADcap (*Hoffberg et al., 2016*). These methods share some basic steps: DNA digestion by one or more restriction enzymes, ligation of sequencing adapters and sample-specific barcodes, followed by optional fragmentation and fragment size selection, multiplexing samples bearing specific molecular tags, i.e. indices and barcodes, and finally sequencing. The sequencing output is thus composed of millions of reads originating from all the targeted homologous loci. The usual bioinformatic steps consist in sample demultiplexing, clustering sequences in loci and identifying informative homologous variations. If a reference genome exists, the most widely used strategy is to align the reads to this reference genome and to perform a classical variant calling, focusing on small variants, Single Nucleotide Polymorphisms (SNPs) and small Insertion-Deletions (INDELs). However, RAD-Seq approaches are used on non-model organisms for which a reference genome does not exist or is poorly assembled. The fact that all reads sequenced from the same locus start and finish exactly at the same position makes it easy to compare directly reads sequenced from a same locus. To de novo build homologous genomic loci and extract informative variations, several methods have been developed, such as *STACKS* (*Catchen et al., 2013*) and *PyRAD* (*Eaton, 2014*), as well as its derived rewritten version *IPyRAD* (*Eaton & Overcast, 2020*), being the most commonly used in the population genomics community.

The main idea behind these approaches is to group reads by sequence similarity into clusters representing each a distinct genomic locus. Since reads originating from the same locus start and end at the same positions, they can be globally aligned, sequence variations can then be easily identified and a consensus sequence is built for each locus. The key challenge is therefore the clustering part. To do so, the classical approach relies on all-vs-all alignments. To reduce the number of alignments to compute, the clustering is first performed within each sample independently, then sample consensuses are compared between samples. Nevertheless the number of alignments to perform remains very large in datasets composed of many large read sets. Importantly, analysis of RAD-Seq data is highly dependent on the chosen clustering method, the sequencing quality and the dataset composition, such as the presence of inter and/or intra-specific specimens or the number of individuals. Thus, existing tools allow customization of numerous parameters to fine-tune the analysis. Particularly, both methods have parameters controlling the granularity of clustering: the number of mismatches allowed between sequences of a same locus within and among samples for *STACKS* and the percentage of similarity for *PyRAD*. These can be arbitrarily fixed by the user, but have a significant impact on downstream analyses (*Shafer et al., 2017*).

We present here *DiscoSnp-RAD*, an utterly different approach to predict de novo small variants (SNPs and indels) from large RAD-Seq datasets, without performing any read clustering, avoiding all-vs-all read comparisons and without relying on a critical similarity threshold parameter. *DiscoSnp-RAD* takes advantage of the *DiscoSnp++* approach (*Uricaru et al., 2015*; *Peterlongo et al., 2017*), that was initially designed for de novo

prediction of small variants, from shotgun sequencing reads, without the need of a reference genome. The basic idea of the method is a careful analysis of the *de Bruijn graph* built from all the input read sets, to identify topological motifs, often called *bubbles*, generated by polymorphisms. Notably, those bubbles arise whatever the global similarity level between homologous reads, explaining why *DiscoSnp-RAD* is free of similarity-related parameters. Note that *STACKS2* also uses a *de Bruijn graph* approach, but in a different way, as it is used to build a so-called "RAD-locus" contig catalog on which reads are aligned for calling SNPs (*Rochette, Rivera-Colón & Catchen, 2019*).

After validation tests on simulated datasets of increasing size, we present an application of the *DiscoSnp-RAD* implementation on double-digest RAD-Seq data (ddRAD) from a genus-wide sampling of parasitic flies belonging to *Chiastocheta* genus. Using *DiscoSnp-RAD*, the 259 individuals analyzed could be assigned to their respective species. Moreover, within-species analyses focused on one of these species, identified variants revealing population structure congruent with sample geographic origins. Thus, the information obtained from variants identified by *DiscoSnp-RAD* can be successfully used for population genomic studies. The main notable difference between *DiscoSnp-RAD* and concurrent algorithms stands in its easiness to use, in the fact that it does not require fine parameter tuning, and in its execution time, as it is substantially faster than *STACKS* and *IPyRAD*.

## MATERIALS AND METHODS

### *DiscoSnp-RAD*: RAD-Seq adaptation of *DiscoSnp++*

Originally, *DiscoSnp++* was designed for finding variants from whole genome sequencing data. To adapt to the RAD-Seq context, the core algorithm of *DiscoSnp++* was extended and modified as shown "*DiscoSnp-RAD*: RAD-Seq adaptation of *DiscoSnp++*" and "Computing Allele Coverage and Inferring Genotypes". Also, as presented "Clustering Variants per Locus" and "Various Optional Filtering Options", specific features for post-processing were added to the whole pipeline.

*DiscoSnp++* **basic algorithm.** We first recall the fundamentals of the *DiscoSnp++* algorithm, which is based on the analysis of the *de Bruijn graph* (dBG) (*Pevzner, Tang & Tesler, 2004*), which is a directed graph where the set of vertices corresponds to the set of words of length $k$ ($k$-mers) contained in the reads, and there is an oriented edge between two $k$-mers, say $s$ and $t$, if they perfectly overlap on $k - 1$ nucleotides, that is to say if the last $k - 1$ suffix of $s$ equals the first $k - 1$ prefix of $t$. In this case, we say that $s$ can be *extended* by the last character of $t$, thus forming a word of size $k + 1$. A node that has more that one predecessor and/or more than one successor is called a branching node. Small variants, such as SNPs and INDELs, generate in the dBG recognizable patterns called "bubbles". A bubble (Fig. 1A) is defined by one *start* branching node that has, two distinct successor nodes. From these two children nodes, two paths exist and merge in a *stop* branching node, which has two predecessors. The type of the variant, whether it is a single isolated SNP, several close SNPs (distant from one another by less than $k$ nucleotides) or an INDEL, determines the length of each of the two paths of the bubble.
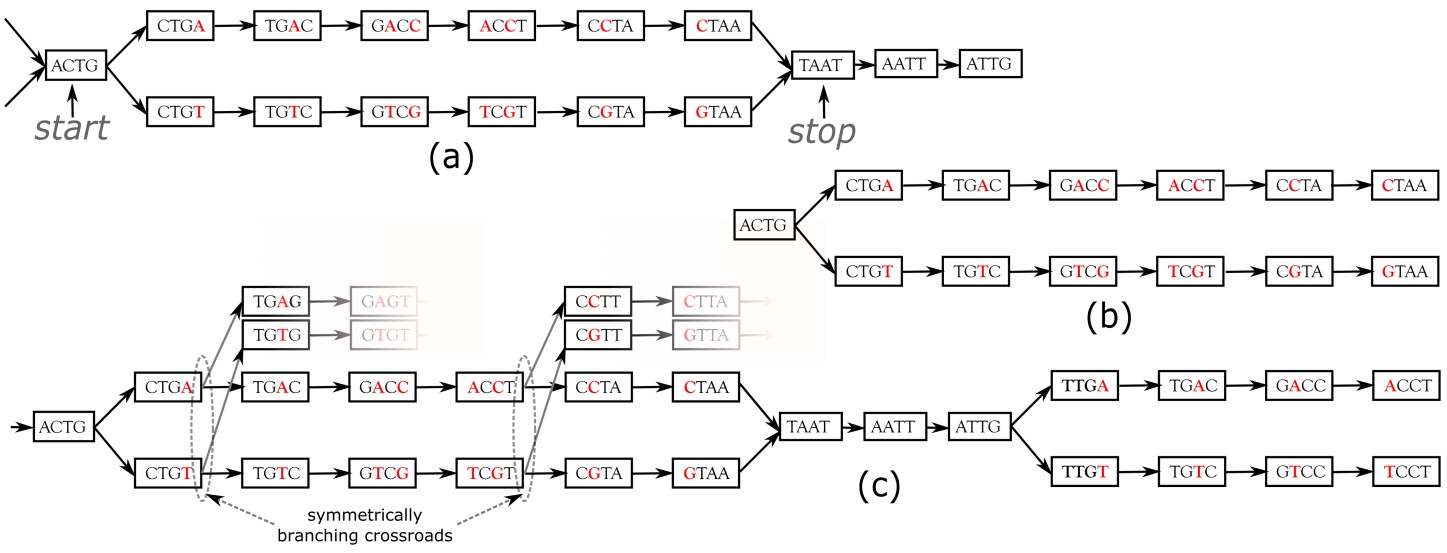

**Figure 1** **Examples of bubbles detected by SNPs in a toy de Bruijn graph, with *k* = 4.** In (A) the bubble is complete: this corresponds to a bubble detected by *DiscoSnp++*. In (B), the bubble is symmetrically truncated: it is composed of a branching node ("*ACTG*") whose two successors lead to two distinct paths that both have the same length and such that their last two nodes have no successor. The graph (C) shows an example of two bubbles from the same locus. The leftmost bubble contains two symmetrically branching crossroads.

*DiscoSnp++* first builds a dBG from all the input read samples combined, and then detects such bubbles. Sequencing errors or approximate repeats also generate bubbles, that can be avoided by filtering out kmers with a too low abundance in the read sets, and by limiting the type or number of branching nodes along the two paths. Detected bubbles are output as pairs of sequences in fasta format. The second main step of *DiscoSnp++* consists in mapping original reads from all samples on these sequences, in order to compute for each variant, its read depth per allele and per read set. From this coverage information, genotypes are inferred and variants are scored. The final output is a VCF file, where each variant is associated to a confidence score (the *rank*) and is genotyped in each read set, thanks to its allele coverages (*Peterlongo et al., 2017*; *Uricaru et al., 2015*).

In *DiscoSnp-RAD*, these two main steps have been modified to adapt to the RAD-seq context and an additional third step has been developed in order to cluster the variants per locus and to output this information in the final VCF file. In short, *DiscoSnp-RAD* (1) constructs the de Bruijn graph and detects bubbles whose topology correspond to SNPs or indels, (2) maps back reads on found bubble sequences, thus assessing the read coverage per allele and per read set, and (3) performs clustering on predicted sequences. Those three steps are described in the three following sections.

### Bubble detection with DiscoSnp-RAD

*A novel RAD-specific bubble model*

In *DiscoSnp++*, variants distant from less than *k* bp from a genomic extremity could not be detected, as associated bubbles do not open and/or close. This effect is negligible in the whole genome sequencing context, however, in the RAD-Seq context, sequenced genomic regions are limited to a hundred or to a few hundreds nucleotides (the read size), and
thus a large amount of variants are likely to be located at the extremities of the loci. For instance, with reads of length 100 bp, and $k = 31$ (which is a usual $k$ value), on average 62% of the variants are located in the first or last $k$ nucleotides of a locus and cannot be detected by *DiscoSnp++*.

In the RAD-Seq context, all reads sequenced from the same locus start and end exactly at the same position. Thus, variants located less than $k$ bp from loci extremities generate what we call *Symmetrically Truncated Bubbles* (Fig. 1B). Such bubbles start with a node which diverges into two distinct paths that do not meet back, such that both of them cannot be extended because of absence of successor and both paths have exactly the same length. Symmetrically, a variant located less that $k$ bp apart from loci start generates a bubble that is right closed, but that starts with two unconnected paths of the same length.

To further increase specificity of the truncated bubble model, we also constrain the last 3-mer of both paths to be identical. Although this prevents the detection of variants as close as 3 bp from a locus extremity, this enables to identify correctly the type of detected variant. Indeed, when the last $L$ nucleotides of two locus sequences are different, several mutation events could have taken place in the genome resulting in the same observed differences: either an indel (of any size) or $L$ successive substitutions or a combination of the two types. When $L$ is small, all events may be equally parsimonious and we prefer to report none of these instead of a wrong one. Note that this does not prevent to detect loci containing such variants as long as there is at least another variant detected in the locus. The value $L$ was set to 3 because it leads to a relatively low loss of recall (6% with reads of length 100), while the probability of observing by chance three successive matches is low ($= 1/4^3 \approx 1.56\%$). Note that this issue is also present in any mapping or clustering based approaches.

The core of the *DiscoSnp-RAD* algorithm SNP bubble detection is sketched in Algorithm 1. The algorithm is intentionally simplified and hides the process enabling to detect SNPs separated by less than $k$ nucleotides and INDELs. The full and detailed algorithm is proposed in the Supplemental Materials. Basically, after the graph construction, we loop over all its branching nodes (line), each branching node is then considered as a potential bubble extremity. The pair of paths that can be generated from this branching node are explored (lines 5 to the end). Notably, the two paths are created simultaneously nucleotide by nucleotide. The extension stops 1/ if the extension is impossible (line 10, if there exists no nucleotide $\alpha$ such that $kmer_1$ and $kmer_2$ can be extended with $\alpha$); or 2/ if the bubble closes (line 11); or 3/ if the bubble is truncated (line 7).

*Dealing with entangled bubbles*

As RAD-Seq data often include a large number of individuals, this is likely that many SNPs are close to each other (separated by less than $k$ nucleotides), and that a large number of distinct haplotypes co-exist. This situation generates bubbles that are imbricated in one another and what we call "Symmetrically Branching Crossroads" (SBCs), as shown in Fig. 1C. SBCs appear when more than one unique character may be used during extension. All possible extensions are explored (line 12) in presence of SBCs. However, we limit the maximal number of traversals of SBCs per bubble to 5 by default (line 14). This value

---

**Algorithm 1** Simplified overview of the DiscoSnp-RAD SNP bubble detection (Indel bubble detection omitted).

1: Create a de Bruijn graph from all (any number ≥1) read set(s)
2: **for** Each right branching $k$-mer in the graph *start* **do**
3:    **for** each couple of successor $kmer_1$, $kmer_2$ of $k$-mer *start* **do**
4:       *nb sym branching*=0
5:       **while** True **do**
6:          Extend $kmer_1$ and $kmer_2$ with α ∈ {A,C,G,T}
7:          **if** Both $kmer_1$ and $kmer_2$ have no successors **then**
8:             **if** last 3 characters from $kmer_1$ and $kmer_2$ are equal **then**
9:                Output bubble and break
10:          **if** Extension is impossible **then** break
11:          **if** $kmer_1 = kmer_2$ **then** Output bubble and break
12:          **if** two or more possible extending nucleotides α **then**
13:             Increase *nb sym* branching
14:             **if** *nb sym_branching* > 5 **then** break
15:             **else** Explore recursively all possible extensions

---

has been chosen as larger values lead toe longer computation time, larger false positive calls (due to repetitive genomic regions), while not changing significantly recall, as shown in the results. Depending on the user choice, we also propose a "high_precision" mode in which bubbles containing one or more SBC(s) are not detected.

### *Computing allele coverage and inferring genotypes*

In this second step, original reads from all samples are mapped on all bubble sequences, in order to provide the read coverage per allele and per read set. Importantly, this mapping step allows non-exact mapping, allowing a high number of substitutions (up to 10 by default), except on the polymorphic positions of the bubble. As shown in results, this choice enables to maximize the sensibility by allowing numerous variations, while maintaining a high precision as no substitution is authorized on variant positions.

These coverage information enables to infer individual genotypes and to assign a score (called *rank*) to each variant enabling to filter out potential false positive variants. Genotypes are inferred only if the total coverage over both alleles is above a *min_depth* threshold (by default 3), using a maximum likelihood strategy with a classical binomial model (*Peterlongo et al., 2017*; *Nielsen et al., 2011*), otherwise the genotype is indicated as missing ("./."). Variants with too many missing genotypes (by default more than 95% of the samples) are filtered out.

Paralogous genomic regions represent a major issue in population genomic analyses as DNA sections arising from duplication events can be aggregated in the same locus and thus, might encompass alleles coming from non orthologous loci. Allele coverage information across many samples can be used to filter out many of such paralog-induced

variants. As the latter tend to occur in all the samples, their allele frequency is thus non discriminant between samples. An efficient scoring scheme, called the *rank* value in *DiscoSnp++*, reflects such discriminant power of variants. First, we define the Phi coefficient of a given variant for a given pair of samples, as $\sqrt{\chi^2/n}$, with $\chi^2$ being the chi-squared statistics computed on the allele read counts contingency table for this pair of samples, and $n$ being the sum of read counts in this table. This is an association measure between two qualitative variables (here allele vs sample) ranging between 0 (no association) and 1 (maximal association). Then, when more than two samples are compared, the rank value is obtained by computing the Phi coefficient of all possible pairs of samples and retaining the maximum value. We have shown in previous work (*Uricaru et al., 2015*; *Peterlongo et al., 2017*) that paralog-induced variants are likely to generate bubbles in the dBG but with very low rank values (<0.4) contrary to most real variants. This filter is particularly effective when many samples are compared, as in the RAD-seq context. Thus, by default, *DiscoSnp-RAD* discards all variants with such low rank values.

Noteworthy, some real variants can also harbor a low rank value: those which are heterozyguous in strictly all the samples. Such variants should be rare when hundreds of samples are considered. Moreover, discarding such variants should not impact on most downstream analyses, such as deciphering population structure or inferring phylogenies, as their results are mainly based on variants which can discriminate the samples. Only the estimation of heterozygosity levels may be affected by removing such real variants: they may be slightly under-estimated, but they would be more dramatically over-estimated without any filtering of paralog-induced variants.

### Clustering variants per locus

During the bubble detection phase, several independent bubbles can be predicted for the same RAD locus. For instance, Fig. 1C shows a toy example of a the dBG graph associated to a locus. In this case, *DiscoSnp-RAD* detects two bubbles, that give no sign of physical proximity. In several population genomics analyses, such proximity information can be useful, such as in population structure analyses, where this is recommended to select only one variant per locus. In order to recover this information of locus membership, we developed a post-processing method to cluster predicted variants per locus.

The method uses the fact that *DiscoSnp-RAD* is parameterized to output bubbles together with their left and right contexts in the graph, which correspond to the paths starting from each extreme node and ending at the first ambiguity (i.e., a node with not exactly one successor). For instance, the leftmost bubble of Fig. 1C is output as sequences ACTG**ACC**TAATtg and ACTG**TCG**TAATtg, where we represent the context sequences in lower case, and rightmost bubble of the same figure is output as sequences taATTG**A**CCT and taATTG**T**CCT.

By definition of these extensions, if a given locus contains several variants, each bubble of this locus extended with its left and right contexts shares at least one $k-1$-mer with
at least one other so extended bubble of the same locus. For instance, the pairs of sequences of the two bubbles shown Fig. 1C share the $k − 1$-mer TAA (among others).

We exploit this property to group all bubbles per locus. For doing so, we create a graph in which a node is a bubble (represented by its pair of sequences including the extensions), and there is an undirected edge between two nodes $N_i$ and $N_j$ if any of the two sequences of $N_i$ shares at least one $k − 1$-mer with any of the two sequences of $N_j$. Those edges are computed using *SRC_linker* (*Marchet et al., 2016*).

Finally, we partition this graph by connected component. Each connected component contains all bubbles for a given locus and this information is reported in the vcf file. By default, clusters containing more than 150 variants are discarded, as they are likely to aggregate paralogous variants from repetitive regions.

### *Various optional filtering options*

The output of *DiscoSnp-RAD* is a VCF file containing predicted variants along with various information, such as their genotypes and allele read counts in all samples, their *rank* value and the cluster ID (locus) they belong to. This enables to apply custom filters at the locus level, as well as any variant level classical RAD-Seq filters (such as the minimal read depth to call a genotype or the minimal minor allele frequency to keep a variant). Several such RAD-seq filtering scripts are provided along with the main program (https://github.com/GATB/DiscoSnp/tree/master/discoSnpRAD/post-processing_scripts).

## Testing environment

The tests were performed on the GenOuest (genouest.org) cluster, on a node composed of 40 Intel Xeon core processors with speed 2.6 GHz and 252 GB of RAM.

## Validation on simulated datasets

Note that all scripts used for simulations and validations are publicly available https://doi.org/10.5281/zenodo.3724518.

### *Simulation protocol*

RAD loci from *Drosophila melanogaster* genome (dm6) were simulated by selecting 150 bp on both sides of 43,848 PstI restriction sites resulting in 87,696 loci. Several populations, each composed of several diploid individuals were simulated as follows. For each simulated population, SNPs were randomly generated at a rate of 1%. A first subset of them (70%) was introduced in all samples from the population and represent shared polymorphism. The rest of these SNPs (30%) where distributed between samples by a random picking of 10% of them and assigned to each sample. For each sample, 10% of the assigned SNPs, shared and sample specific, are introduced in only one of the homologous chromosomes to simulate heterozygosity. This process was repeated to generate from 5 to 50 populations each composed of 20 individuals. Finally, between 2,109,900 SNPs for 100 samples, and 2,547,337 SNPs for 1,000 samples, were generated. Forward 150 bp reads were simulated on right and left loci, with 1% sequencing errors, with 20× coverage per individual (the complete pipeline is given in Fig. S1).

### Evaluation protocol

For estimating the result quality, predicted variants were localized on the *D. melanogaster* genome and output in a vcf file. To do so, we used the standard protocol of *DiscoSnp++* when a reference genome is provided, using BWA-mem (*Li & Durbin, 2009*). The predicted vcf was compared to the vcf storing simulated variant positions to compute the amount of common variants (true positive or TP), predicted but not simulated variants (false positive or FP) and simulated but not predicted variants (false negative or FN). Recall is then defined as $\frac{\#TP}{\#TP+\#FN}$, and precision as $\frac{\#TP}{\#TP+\#FP}$.

### Comparison with other tools

For comparisons, *STACKS* v2.4 and *IPyRAD* v0.7.30 were run on the simulated datasets. *STACKS* stacks were generated de novo ( denovo_map.pl ), with a minimum of 3 reads to consider a stack (-m 3). On the simulated dataset composed of 100 samples, five values of the parameter -M governing stack merging (i.e., 4, 6, 8, 10 and 12) were tested. On the remaining datasets, the parameter -M was fixed to 6 following r80 method (*Paris, Stevens & Catchen, 2017*). All other parameters were set to default values. Similarly, *IPyRAD* was run using five values of clustering threshold on the dataset composed of 100 samples (i.e., 0.75, 0.80, 0.85, 0.90 and 0.95) and then fixed to 0.80, following r80 method (*Paris, Stevens & Catchen, 2017*), for larger datasets. The other parameters have been kept at the default values. Then, de novo tags from *STACKS* and loci from *IPyRAD* were mapped to the *D. melanogaster* genome using BWA-mem and variant positions were transposed on the genome positions with a custom script.

## Application to real data from *Chiastocheta* species
### Data origin

Tests on real data were performed on ddRAD reads previously obtained for the phylogenetic study of seed parasitic pollinators from the genus *Chiastocheta* (Diptera: Anthomyiidae). The dataset corresponds to the sequencing of 259 individuals sampled from 51 European localities generated by Lausanne University, Switzerland (*Suchan et al., 2017*) (https://www.ebi.ac.uk/ena/data/view/PRJEB23593). A total of 608,367,380 reads were used for the study with an average of 2.3 Million reads per individual.

### Variant prediction and filtering

*DiscoSnp-RAD* was run with default parameters, searching for at most five variants per bubble. For *IPyRAD* the same parameters as in the *Suchan et al. (2017)* study have been applied including a percentage of identity of 75% for the clustering and a minimum coverage of 6. For *STACKS* we applied a minimum coverage by stack (-m) of 3, a maximum number of mismatches allowed among sample (-M) of 8 and a maximum number of mismatches allowed between sample (-n) of 8. On the output vcf from each tools, downstream classical filters were applied to follow as much as possible the filters used in the *Suchan et al. (2017)* study: a minimum genotype coverage of 6, a minimal minor allele frequency of 0.01 and a minimum of 60% of the samples with a non missing genotype for each variant. These filters remove less informative variants or those with an allele specific to a very small subset of samples. These filters were also applied at the

intra-specific level in one of the seven sampled *Chiastocheta* species, that is, *C. lophota*, on the same *DiscoSnp++* output.

### Population genomic analyses

The species genetic structure was inferred using STRUCTURE (*Pritchard, Stephens & Donnelly, 2000*) v2.3.4. This approach requires unlinked markers, thus only one variant by locus, randomly selected, has been kept. The STRUCTURE analysis was carried on the datasets generated by each tool. Simulations were performed with genetic cluster number ($K$) set from 1 to 10. Best $K$ was identified using *Evanno, Regnaut & Goudet (2005)* method. We used 20,000 MCMC iterations after a burn-in period of 10,000. The output is the posterior probability of each sample to belong to each of the possible clusters. For *C. lophota* species, a multivariate analysis were used to investigate intra-specific genetic structure using adegenet R package (*Jombart, 2008*).

### Phylogenomic analyses

Maximum likelihood (ML) phylogenetic reconstruction was performed on a whole concatenated SNP dataset using GTRGAMMA model with the acquisition bias correction (*Leaché et al., 2015*). We applied rapid Bootstrap analysis with the extended majority-rule consensus tree stopping criterion and search for best-scoring ML tree in one run, followed by ML search, as implemented in RAxML v8.2.11 (*Stamatakis, 2014*).

## RESULTS

### Results on simulated data

*DiscoSnp-RAD* was first run on several simulated RAD-Seq datasets composed of an increasing number of samples (from 100 to 1,000) in order to validate the approach, to evaluate its speed and efficiency and to compare it with the other clustering approaches. This experiment shows that *DiscoSnp-RAD* predictions are accurate with a good compromise between recall and precision (see Fig. 2).

On average, 84.6% of the simulated variants are recovered with very few false positive calls, that is, reaching a precision of 98.5% on average. Importantly, these performances are not impacted by the number of input samples in the dataset. For instance, recall varies from 84.6% to 83.3% between the smallest and the largest datasets (100 vs 1.000 samples), and precision from 98.1% to 98.5%.

By comparison with other tools, for each of the tested population sizes, recall and precision are comparable between tools, with typically a recall lower than *STACKS* and *IPyRAD* and an intermediate precision, lower than *IPyRAD* and higher than *STACKS*. The loss of recall may be explained by the fact that *DiscoSnp-RAD* voluntary does not detect the variants within 3 bp of each locus end (see "Methods"). The amount of predicted loci are similar between all tools (Table S2). The main differences between the tools concern the run time and the disk space usage. These differences increase with the number of samples in the dataset. For instance, on the largest dataset composed of 1,000 samples *DiscoSnp-RAD* is more than 3 times faster than *STACKS* and more than 5 times faster than *IPyRAD*. Moreover, if we consider the cumulative time required to test different

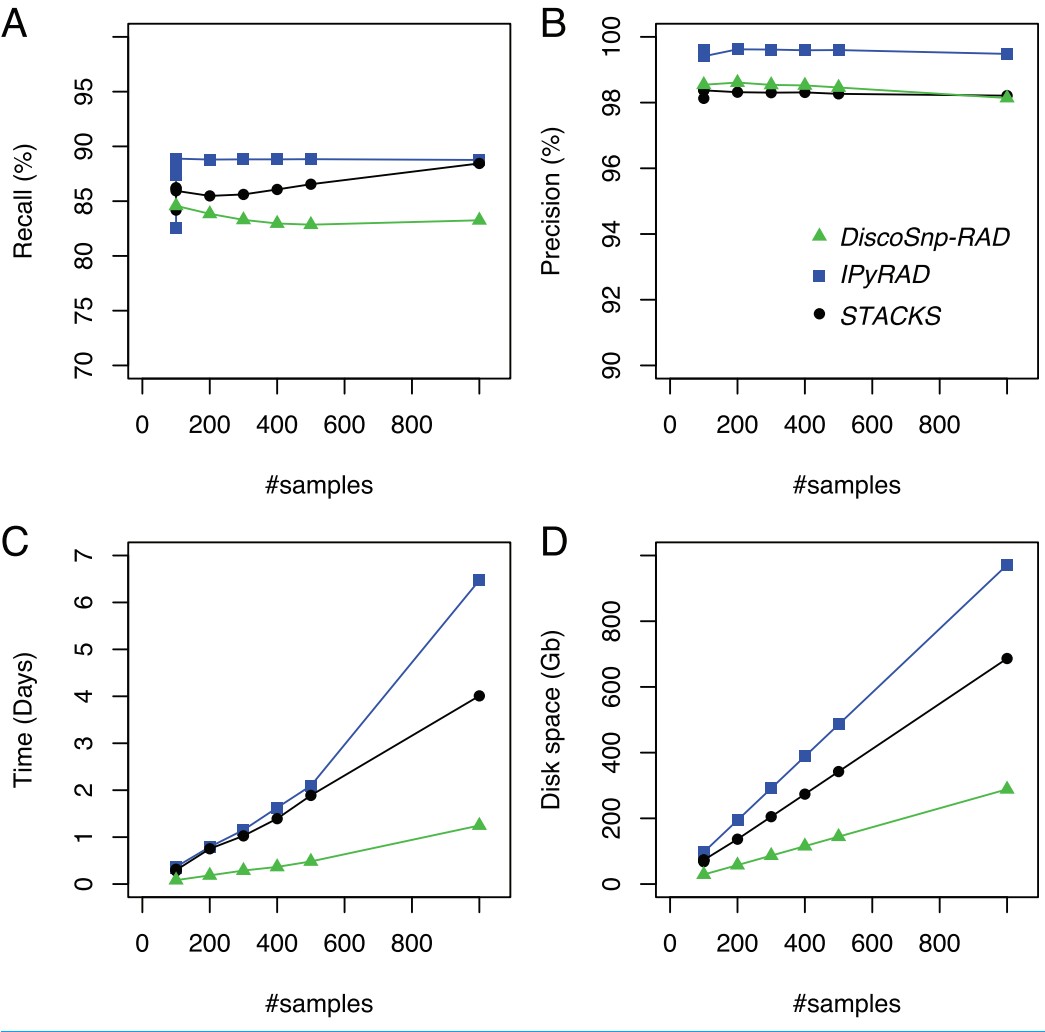

**Figure 2 Recall (A), precision (B), time (C) and space (D) evolution on simulated data with different sampling sizes.** For the sampling of 100 samples, five parameter sets were tested for *IPyRAD* and *STACKS* (see "Material and Methods" for details).

parameters for *STACKS* and *IPyRAD*, that is, five sets of parameters for each tool, *DiscoSnp-RAD*, without parameter setting is more than 15 times faster than *STACKS* and more than 25 times faster than *IPyRAD*. Regarding the disk space used by the tool during the process, *DiscoSnp-RAD* requires only a small amount of space compared to the other tools. Full RAM memory, disk usage, and computation times of *DiscoSnp-RAD* are provided in Table S1.

### Robustness with respect to parameters

In *DiscoSnp-RAD*, the main parameter is the size of *k*-mers, used for building the dBG. As shown Fig. 3, *DiscoSnp-RAD* results are robust with respect to *k*, the main parameter, and its fine choice is thus not crucial. This figure also highlights the results robustness with respect to other parameters such as the maximal number of predicted SNPs per bubble (5 by default), the maximal number of substitutions authorized when mapping reads on bubble sequences (10 by default), and the maximal number of symmetrically

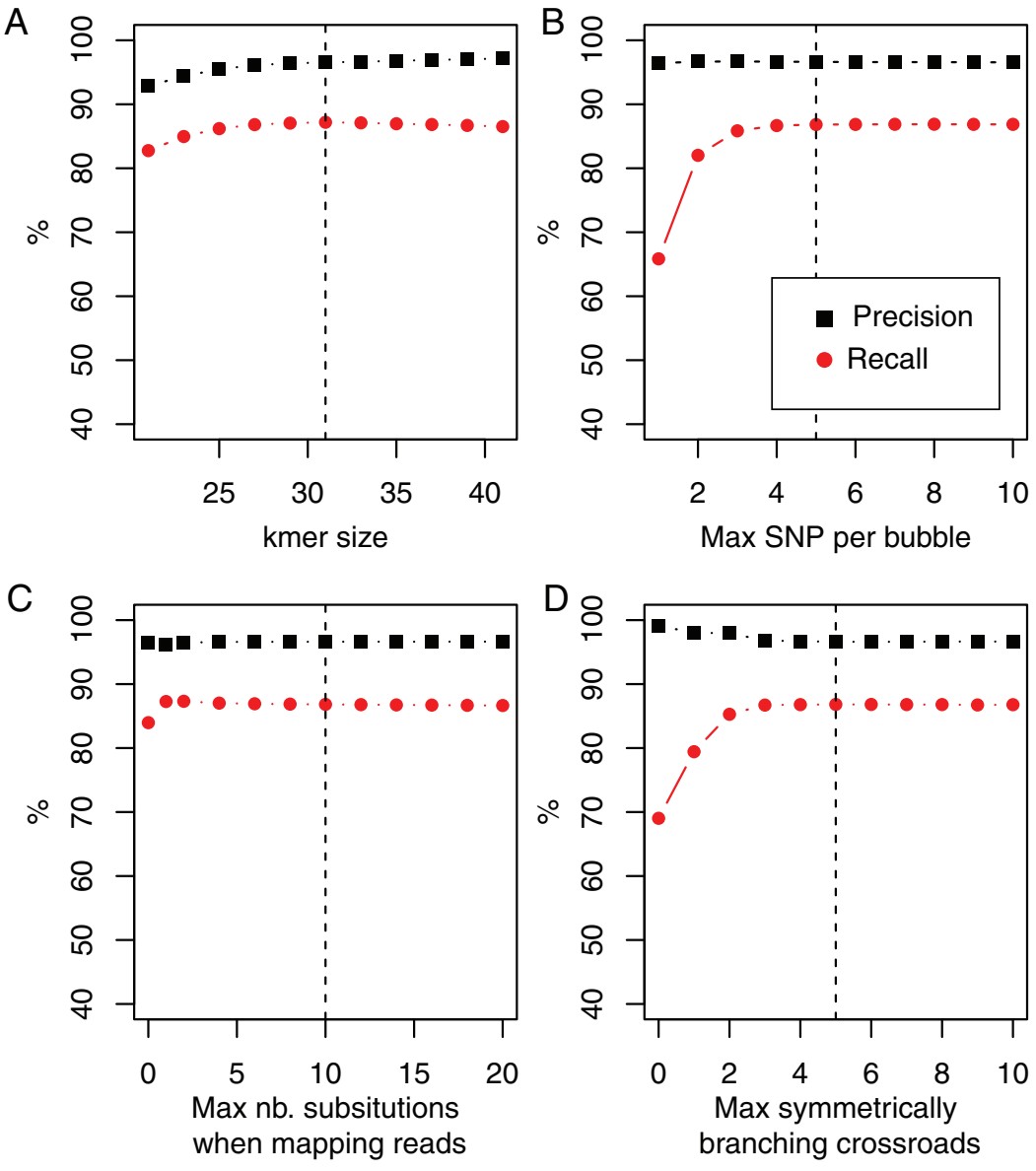

**Figure 3** **Recall and precision on simulated data of 100 samples using DiscoSnp-RAD with respect to (A).** *k*-mer sizes, (B) maximal number of authorized SNP per bubble, (C) maximal number of authorized substitutions while mapping reads on predicted variants sequences, and (D) maximal number of symmetrically branching crossroads. Dashed vertical line represents on each plot the chosen default value.

branching crossroads (also 5 by default). Concerning this last parameter, Fig. 3 also shows the advantages of the "high_precision" mode which sets this parameter to zero, leading to a precision of nearly 100%.

We enriched results presented in this figure with two additional simulated datasets, following the protocol presented in the "Simulation protocol" section, but introducing SNPs at rates of 0.5% and 2%, instead of 1%. Results are presented in the Fig. S2. They also highlight the robustness of results and the rational for the default parameter choices with two times more and two times less simulated diversity.

The robustness of *DiscoSnp-RAD* is an important point as other state-of-the-art tools are extremely sensible to their parameters, especially those directly linked to the expected sequence divergence, and require time consuming processes to set them properly (*Shafer et al., 2017*).

## Results on real data

In this section, we present an application of the *DiscoSnp-RAD* implementation on ddRAD sequences obtained from the anthomyiid flies from the *Chiastocheta* genus. In this genus, classical mitochondrial markers are not suitable for discriminating the morphologically described species (*Espíndola, Buerki & Alvarez, 2012*). Although RAD-sequencing dataset phylogenies supported the species assignment (*Suchan et al., 2017*), the interspecific relationships between the taxa could not be resolved with high confidence due to high levels of incongruences in gene trees (*Gori et al., 2016*; *Suchan et al., 2017*). The dataset is composed of 259 sequenced individuals from 7 species. Results obtained on *DiscoSnp-RAD* were compared to the prior work of Suchan and colleagues, based on *pyRAD* analysis (*Suchan et al., 2017*). In addition, we provide a performance benchmark of *STACKS*, *IPyRAD* and *DiscoSnp-RAD* run on this dataset.

### Recovering all Chiastocheta species

Variant calling was run on the 259 *Chiastocheta* samples with *DiscoSnp-RAD*. Before filtering, 115,920 SNPs were identified. After filtering, 4,364 SNPs, located in 1,970 clusters, were retained and used for population genomic analyses. The total number of clusters is coherent with the 1,672 loci from *Suchan et al. (2017)*.

Then, following the requirements of the STRUCTURE algorithm, only one variant per cluster was retained, resulting in a dataset composed of 1,970 SNPs. The most likely value of $K$ is 7 (Fig. S3) and corresponds to the seven species described in *Suchan et al. (2017)*. STRUCTURE successfully assigned samples to the seven species, consistent with the morphological species assignment and previously published results (*Suchan et al., 2017*) (Fig. 4). The assignment values represent the probability with which STRUCTURE assigns a sample to a cluster, depending on the information carried by the variants. The assignment values are high with an average of 0.992 (sd 0.022) across samples and a minimum assignment of 0.810. These values are comparable to the assignment values obtained by *Suchan et al. (2017)* with an average of 0.977 (sd 0.042) and a minimum of 0.685. Genetic structure has also been investigated for the two other tools and give very similar population assignations (Fig. S4).

The phylogeny realized with RAxML on the 4,364 SNPs obtained after filtering, identifies clearly seven clusters representative of the seven species which are coherent with the clusters obtained by *Suchan et al. (2017)* (Fig. 4). The internal branches separating the seven species are well supported by high bootstrap values.

### Recovering phylogeographic patterns

To assess the utility of *DiscoSnp-RAD* results for investigating the intra-specific genetic structure, we then focused the analysis on 40 samples of *C. lophota* species. From the same

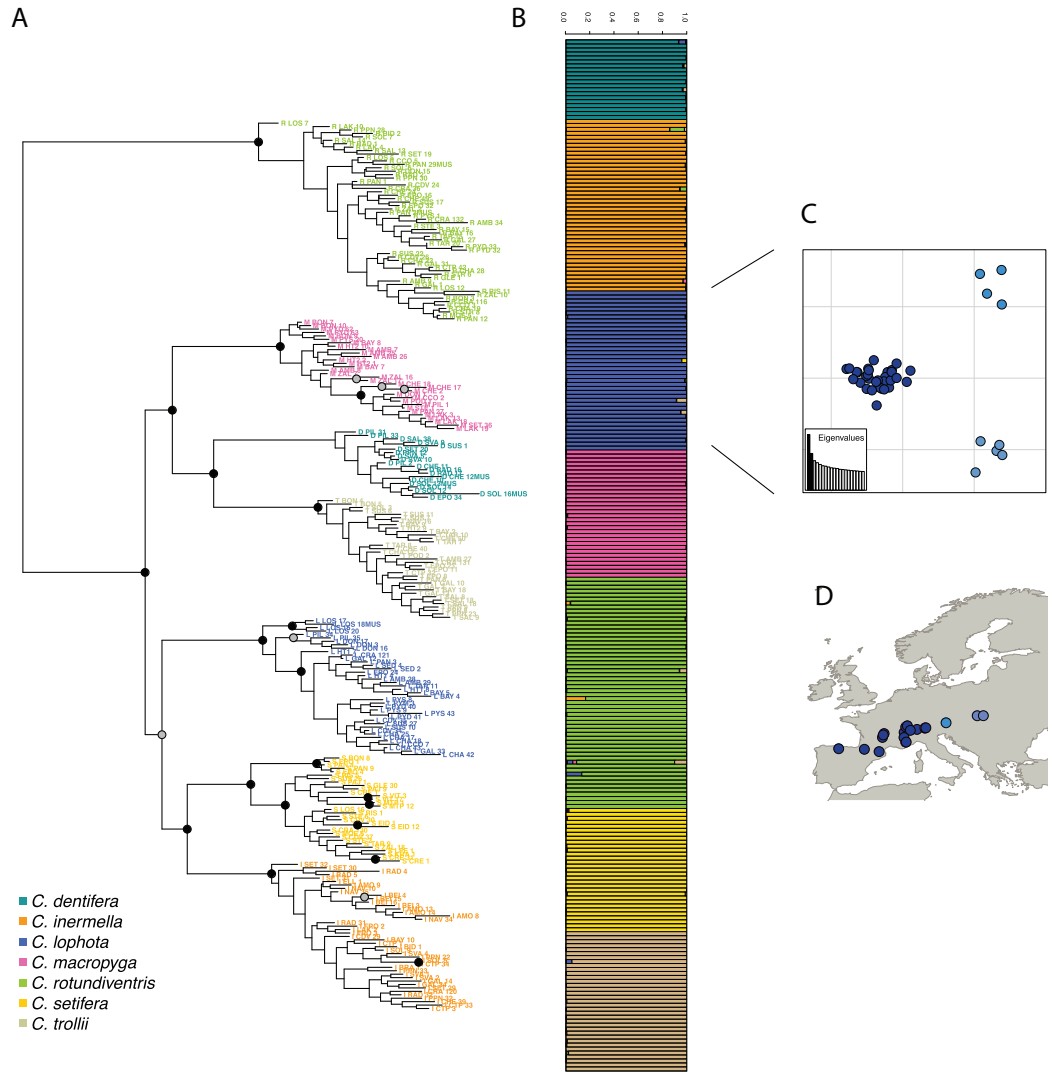

- ■ *C. dentifera*
- ■ *C. inermella*
- ■ *C. lophota*
- ■ *C. macropyga*
- ■ *C. rotundiventris*
- ■ *C. setifera*
- ■ *C. trollii*

**Figure 4** **(A) RAxML phylogeny realized on all variants predicted by *DiscoSnp-RAD*.** Bootstrap node supports > 80 are shown denoted by gray dots, bootstrap node supports > 90 are shown denoted by black dots. (B) STRUCTURE results obtained with SNP only and all variants on the seven Chiastocheta species. (C) Plot of the two first PC from a multivariate analysis on *C. lophota* samples and (D) their geographic distribution (figure made with *Natural Earth Contributors (2020)*).

vcf file obtained with the 259 samples, the 40 *C. lophota* samples were extracted and the same filters, that is, MAF, missing etc., were applied on this *C. lophota* dataset.

We obtained 1,306 SNPs by selecting one variant per locus extracted from 4,364 variants identified in this species. The multivariate analysis of this dataset identify three populations comprising respectively 31, 5 and 4 samples (Fig. 4). Notably, the genetic structure follows the geographic distribution of the samples, with samples from one population originating from western locations, another population from eastern locations and an intermediate population. Geographic structuring is the most frequent structuration factor observed in population genetics, pointing to the geographical isolation

of divergent lineages. This clear geographic structuring is another hint that *DiscoSnp-RAD* recovers real biogeographic signal.

### Breakthrough in running time

*DiscoSnp-RAD* run on the 259 *Chiastocheta* samples took about 30 h. This comprises the whole process from building the dBG to obtaining the final filtered vcf file with 1 SNP per locus. To compare the *DiscoSnp-RAD* performances with *STACKS* and *IPyRAD* on real data, we ran each of these tools using default parameters on the 259 Chiastocheta samples and measured running time and maximum memory usage. The difference is remarkable, *DiscoSnp-RAD* is more than 4.65 times faster than *STACKS* (running time 138 h) and 2.8 times faster than *IPyRAD* (running time 82 h) to perform the whole process. Moreover, contrary to *DiscoSnp-RAD*, *STACKS* and *IPyRAD* should be run several times to explore the parameters which represent a considerable amount of time and memory. For instance, in *Suchan et al. (2017)*, *IPyRAD* was run with 5 different combinations of parameter values, *DiscoSnp-RAD* being thus 14 times faster than *IPyRAD*.

## DISCUSSION

### *DiscoSnp-RAD* efficiency

*DiscoSnp-RAD* produced relevant results on ddRAD data from *Chiastocheta* species. SNPs identified allowed us to successfully (i) distinguish the seven species based on the STRUCTURE algorithm, and (ii) reconstruct the phylogenetic tree of the genus, coherent with the phylogenies previously published (*Suchan et al., 2017*). Moreover, on the intra-specific scale, we obtained geographically meaningful results within *C. lophota* species. The variants identified by *DiscoSnp-RAD* can be used to study the species or population genetic structure and could be used to investigate deeper the mechanisms at the origin of this structure such as potential gene flow between populations or their demographic histories. In addition, *DiscoSnp-RAD* is also able to identify INDELS (*Peterlongo et al., 2017*). They were not used in this study but are available for users.

Furthermore, the use of *DiscoSnp-RAD* presented considerable advantages in the run-time, and parameters choice, compared to other common de novo RAD analysis tools, as described below.

### Run-time

The use of *DiscoSnp-RAD* dramatically decreased the overall time for discovering and selecting relevant variants, as compared to other tools. This is made possible thanks to the use of a unique indexing data structure, the dBG built from all the reads. To build this graph, reads do not need to be compared to each other. *DiscoSnp-RAD* speed depends on the graph size and at a lesser extend on the number of reads. Importantly, it is not expected to increase quadratically with the dataset size.This can likely be anticipated that with the drop of sequencing costs, RAD projects will grow in size, either by using higher frequency cutting enzymes to obtain a dense genome screening, by increasing the sequencing depth to compensate sequencing variation or by increasing the number of samples. In this context, *DiscoSnp-RAD* will more easily scale to such very large datasets.

### Easy parameter choice

Another substantial advantage of using *DiscoSnp-RAD* is the fact that parameters are not directly linked to the level of expected divergence of the compared samples. In fact, they impact the number and type of detected variants, but are not related to the subsequent clustering step. As a result, same parameters can be used whatever the type of analysis (e.g., intra or inter-specific), contrasting with classical tools in which parameters govern loci recovering. Indeed, in *STACKS*, the parameters governing the merge of the stacks can compromise the detection of relevant variants if they are not adapted to the studied dataset (*Shafer et al., 2017*). Therefore, the authors recommend to perform an exploration of the parameter space before downstream analyses (*Paris, Stevens & Catchen, 2017*). This is extremely time consuming, up to 1 month as confessed by the authors *Rochette & Catchen (2017)*, and may not always result in interpretable conclusions. In *IPyRAD*, the similarity parameter for clustering also impacts variant detection, and usually several values have to be tested to choose the best, as exemplified by *Suchan et al. (2017)* who tested five different values.

### By-locus assembly

*DiscoSnp-RAD* output is a vcf file including pseudo-loci information, that allows the application of standard variant filtering pipelines. One next objective is to recover loci consensus sequences, that could be used for phylogenetic analysis based on full locus sequences. This could be achieved by performing local assemblies per individual, from all bubbles contained in a cluster.

### Sequencing error rate

Our tests on simulated data sets, performed with 1% error rate, show that, in this worst case scenario, *DiscoSnp-RAD* can deal with a high error rate and can thus afford analyses of data not generated with the most recent sequencing technologies. With a lower error rate, the good performances of *DiscoSnp-RAD* are confirmed as shown by the results obtained on the real *Chiastocheta* dataset. The breakthrough in running time with respect to the other approaches is slightly reduced with the real dataset and this could be due to *STACKS* and *IPyRAD* being more impacted by sequencing errors in the data.

### Potential applications

*DiscoSnp-RAD* can handle all types of RAD data including original RAD-Seq, GBS, ddRAD, etc. In addition it is able to use reads 2 from original RAD-Seq data that are often difficult to analyze. These reads do not start and finish at the same position. Properly recovery of loci is therefore not possible with read stacking approaches. This problem does not exist when using *DiscoSnp-RAD*, and variants present in reads 2 can also be called. Indeed, the *DiscoSnp-RAD* method, being not based on stacks of reads, is able to detect any variants that generate bubble motifs in the dBG, thus even if present in reads whose starting positions differ. More precisely, if in a given locus some reads from reads 1 overlap some reads from reads 2 over at least $k - 1$ characters, then all variants from this locus

are clustered together and hence detected as belonging to the same locus. Conversely, variants detected from reads 1 are not related to variants detected from reads 2.

This ability of *DiscoSnp-RAD* to handle reads that do not necessary start at the same genomic position makes it particularly well suited to analyze the datasets produced by another group of genome-reduction techniques, namely sequence capture approaches (*Grover, Salmon & Wendel, 2012*). In these techniques, DNA shotgun libraries are subject to enrichment using short commercially-synthesized (*Faircloth et al., 2012*) or in-house made (*Suchan et al., 2016*) DNA or RNA fragments acting as "molecular baits", that hybridize and allow separation of homologous fragments from genomic libraries. One of such promising approaches is HyRAD, a RAD approach combining the molecular probes generated using ddRAD technique and targeted capture sequencing, designed for studying old and/or poor quality DNA, likely to be too fragmented for RAD-sequencing (*Suchan et al., 2016*). In HyRAD, capturing randomly fragmented DNA results in reads not strictly aligned and covering larger genomic regions than RAD-Seq. Therefore RAD tools can not be used to reconstruct such loci, and the current analysis consists in building loci consensuses from reads, and then calling variants by mapping back the reads on it. The use of *DiscoSnp-RAD* should simplify this process in a single de novo calling step, well adapted to the specificities of data generated by reduction approaches: many compared samples, high polymorphism and clustering by locus.

## CONCLUSION

We propose *DiscoSnp-RAD*, an original method dedicated to the *de-novo* analyze of RAD-Seq data. We have shown that on simulated data, the quality of the results is comparable to those obtained by state-of-the art tools, *STACKS* and *IPyRAD*. On real data, *DiscoSnp-RAD* provides relevant results, enabling the structuring at inter- and intra-level species, accurate enough for recovering the phylogeographic patterns.

Due to its methodological approach which is utterly different from existing methods, *DiscoSnp-RAD* drastically reduces computation times and memory requirements. Another key difference stands in the fact that *DiscoSnp-RAD* does not rely on fine tuning of parameters, contrary to existing methods that rely on critical parameters, as those related to the input sequence similarity.

## ACKNOWLEDGEMENTS

The authors thank Camille Marchet for her precious help on the clustering implementation. Computations have been made possible thanks to the resources of the Genouest infrastructures.

### Funding

This work was supported by the French ANR-14-CE02-0011 SPECREP grant. The funders had no role in study design, data collection and analysis, decision to publish, or preparation of the manuscript.

## Grant Disclosures

The following grant information was disclosed by the authors:
French SPECREP: ANR-14-CE02-0011.

## Competing Interests

The authors declare that they have no competing interests.

## Author Contributions

- Jérémy Gauthier conceived and designed the experiments, performed the experiments, analyzed the data, prepared figures and/or tables, authored or reviewed drafts of the paper, and approved the final draft.
- Charlotte Mouden performed the experiments, analyzed the data, authored or reviewed drafts of the paper, code implementation, and approved the final draft.
- Tomasz Suchan performed the experiments, analyzed the data, authored or reviewed drafts of the paper, and approved the final draft.
- Nadir Alvarez performed the experiments, analyzed the data, authored or reviewed drafts of the paper, and approved the final draft.
- Nils Arrigo performed the experiments, analyzed the data, authored or reviewed drafts of the paper, and approved the final draft.
- Chloé Riou conceived and designed the experiments, performed the experiments, authored or reviewed drafts of the paper, code implementation, and approved the final draft.
- Claire Lemaitre conceived and designed the experiments, performed the experiments, analyzed the data, prepared figures and/or tables, authored or reviewed drafts of the paper, code implementation, and approved the final draft.
- Pierre Peterlongo conceived and designed the experiments, performed the experiments, analyzed the data, prepared figures and/or tables, authored or reviewed drafts of the paper, code implementation, and approved the final draft.

## Data Availability

Code and additional information are available here:
https://github.com/GATB/DiscoSnp.

All validation scripts used for our tests are available at Zenodo: Gauthier, J (2020). "Custom scripts for reproducing DiscoSnp-RAD experiments". Zenodo. Script.
DOI 10.5281/zenodo.3724518.

## Supplemental Information

Supplemental information for this article can be found online at http://dx.doi.org/10.7717/peerj.9291#supplemental-information.

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
