# Peer review of "DiscoSnp-RAD: de novo detection of small variants for RAD-Seq population genomics"

_PeerJ, doi:10.7717/peerj.9291_

## Round 0.1 · original submission · Major Revisions

Dear authors,

I have received three reviews. All are positive, but all reviewers requested revisions. I agree with the reviews, and I myself would like to see a number of issues addressed:

1) Make your method more flexible, i.e. there are a number of hard wired parameter defaults that should not be hard wired. Keep your default values, but make it easy to change them by passing alternate values to the program.
2) It would be good to provide the scripts that were used to run the benchmarking simulations.
3) It will be import to make a clear documentation or a cookbook example that potential users can run through.
4) I would suggest that you provide a script that will join forward and reverse reads, so that one can use information from both reads. This may not be very relevant for RAD data, but given that homologous fragments have same size in ddRAD data, this would permit capture of more information. Again this could be implemented through a parameter value that is passed onto the program.

Otherwise, I think that the potential time savings of DiscoSNP-RAD have the potential to make it a widely used tool, but its acceptance will also depend on its ease of use, i.e. clear documentation and cookbook examples, and that the program runs out-of-the-box.

I look forward to seeing your revision.

Tomas Hrbek

Reviewer 1 ·

Basic reporting

Literature cited and background in the introduction are sufficient. There are some minor grammar issues throughout, which I note in the comments section below. As an example, the caption for figure 1 states: "The leftmost bubble contains two symmetrically crossroads.", yet I believe this should be 'symmetrically branching crossroads'. Another pass through the manuscript with an eye for things like this will be beneficial.

I rarely find pseudo-code to be informative, so I would recommend removing Algorithm 1.

Throughout the manuscript the sequence data for any given sample is often referred to as a 'dataset', which is pretty non-standard. For example (line 179): "...allele read counts contingency table over all possible pairs of datasets…". In my mind a dataset is the totality of all data for a given project, i.e. all the sequence data for all the samples.

Line 184: Are there other things besides "approximate repeats" that might generate a low rank value for real variants, which might thus be excluded unnecessarily?

Experimental design

The research is within the aims and scope of the journal, and it is well defined and relevant. Certainly providing a high-performance method to extract variants from raw RADSeq data is a worthy target. Overall, the structure of the materials and methods section looks good and makes sense, though there are some sections where critical information is missing, which I highlight below.

Section 2.4 Application to real data from Chiastocheta species - This section doesn't provide sufficient information to reproduce many parts of the analysis, especially the stacks/ipyrad assemblies and structure analyses.

For paired end data, if R1 and R2 are non-overlapping then they will be treated as independent reads, correct? This discards valuable information about linkage between snps from R1 and R2.

Line 78: Stacks2 makes use of de Bruijn graphs as well, so it might be nice to highlight the differences of this method versus stacks2.

Line 136: "Although this prevents the detection of variants as close as 3 bp…." If I'm not mistaken, this means that "loci" with variants in the last 3bp are discarded. Is there some other way this could be handled w/o discarding these real variants?

Line 162: "(data not shown)". Please show this data, or at least indicate what constitutes the threshold for significant change in recall.

Line 179: The explanation of the rank value could be clarified a bit. I understand what it's doing now, but I had to look at the code, look up Phi coefficient, and read this paragraph several times before I got it.

Line 197: "... each bubble of this locus should share one k −1-mer…", but what if they do not? "Should" is not a very strong condition, and it would seem that if the bubbles from the same locus do not share a k-1-mer then they will be treated as independent loci, if I'm not wrong.

Line 227: The 1% simulated sequencing error is substantially higher than is typical of current Illumina instruments, which tend more toward 0.1%.

Line 229: The evaluation protocol should specify whether multiple snps from a given locus are identified as coming from the same locus, or if they come from different loci. It's true that the precision is quite good, but if the snps are being split up into individual loci per snp then it's losing information about linkage.

Line 242: It's not clear why the denovo tags/loci from stacks/ipyrad were mapped to the D. melanogaster genome in different ways.

Line 249: The "custom script" here is not provided in the github repository.

Line 332: Recovering phylogeographic patterns - It's more typical to run STRUCTURE analysis for various values of K and to assess the likelihood of each K value. It's well known that the current methods of evaluating likelihoods of K values are unable to evaluate the likelihood of K=1, so choosing K=2 here seems somewhat arbitrary, especially given the very small ancestral components assigned to the few Eastern samples.

Validity of the findings

One of the big selling points of this method appears to be the ability to run without tuning parameters, yet in practice I found that `run_discoSnpRad.sh` has many parameters that are configurable, and that changes in these have marked impacts on the resulting outputs. It's also not clear how many of the parameters of the component parts (dbgh5, kissnp2, kissreads2, fasta_and_cluster_to_filtered_vcf.py) are configurable and how many are simply hard coded. It seems like a mix of these, as far as I can tell. Line 294: "A major advantage of DiscoSnp-RAD stands in the fact that it does not require fine parameter tuning." In examining the code in the file run_discoSnpRad.sh (provided in the github repository), in the section under DEFAULT VALUES, it appears there are at least 10 parameters that roughly correspond to very similar parameters in ipyrad and stacks (e.g. min and max coverage, max number of snps per bubble, etc). I would say, rather than not requiring fine tuning, that DiscoSnp-RAD tries to choose sensible defaults. Figure 3 shows somewhat similar results for variation of the k-mer size parameter, but in order to make the general claim that discoSnpRad doesn't need parameter tuning I think a little more work needs to be done to show that the various parameters do not change the results.

Line 325: Unless I'm mistaken, the topology of the RAxML tree in figure 4 and the topology of the ML tree genreated from RADSeq data in Suchan et al 2017 (Figure 1b) actually do _not_ agree. In the manuscript (C. macropyga, (C. Trolli, C. dentifera)) is reciprocally monophyletic with respect to (C. lophota, (C. setifera, C. inermella)), but in Suchan this is not true.

Additional comments

Line 29: "... due to his completely different principle…", I would say "its" instead of "his".

Line 56: Add or update citations to include the most recent versions of stacks and ipyrad:

Rochette, N. C., Rivera‐Colón, A. G., & Catchen, J. M. (2019). Stacks 2: Analytical methods for paired‐end sequencing improve RADseq‐based population genomics. Molecular ecology, 28(21), 4737-4754.

Eaton, D. A., & Overcast, I. (2020). ipyrad: Interactive assembly and analysis of RADseq datasets. Bioinformatics.

Line 71: "arbitrarily"

Line 82: "Chiastocheta species". Chiastocheta is the genus, correct?

Line 95: "Graph" is not a proper noun, so it should not be capitalized, in my view. "de Bruijn graph" is better. This is a small thing. It's correct on line 78.

Line ~105: Figure 1 caption: "symmetrically" should be "symmetrical".

Line 108: "... filtering out kmers with a too low abundance in the read sets, and by limiting the type or number of branching nodes along the two paths." These seem like parameters that could take different values based on characteristics of the specific dataset in hand. How are "Low abundance" and "type or number of branches allowed" defined by DiscoSnp++? Are these exposed to the user? If not, how are the defaults chosen?

Line 159: "We limit the maximal number of traversals of SBCs per bubble to 5." This is another example of a parameter that is simply hard coded, rather than allowed to be adjusted by the user based on their understanding of the data. It is later stated that "This value has been chosen after tests showing that larger values lead to longer computation…" and so on, but there is no indication of the datasets used for these tests. Were they from simulated or real data? If from real data, what was the taxonomic scope of investigation? Within species genetic diversity could have a drastic impact on setting of this parameter.

Line 167: "...allowing up to 10 substitutions...", again, this seems to be an arbitrarily chosen and fixed value. Stacks and ipyrad get a lot of attention in the introduction based on their parameterizations, yet this method seems to be simply making these parameter decisions for the end user.

Line 175: "...deriving from coalescent events." I would prefer "alleles sampled from non-homologous loci", or something like this. If the alleles are paralagous, they would certainly share a common ancestor, just deeper in time.

Line 185: "2.1.3 Clustering variants per loci", should be 'locus'.

Lines 173-184: There is a lot in this paragraph that is unclear. For example, "approximated repeats" are not well defined. "...read counts contingency table over all possible pairs of datasets…", it's not clear which pairs of datasets are referred to here.

Line 203: "Clusters containing more than 150 variants are discarded…." This is another hard coded parameter of the model.

Line 240: ipyrad clustering values tested on the simulated data appear not to have been included, "... composed of 100 samples (ie. )...". Also, how final stacks `-M` and ipyrad clustering threshold values were chosen is not clear.

Line 262: STRUCTURE is not simulation based.

Line 318: This is a somewhat informal description of how STRUCTURE works, it's also more "methods-y", in my mind and so doesn't belong in the results. I would move to the methods and make it more formal.

Line 382-384: It is not true that Stacks or ipyrad only use R2 mainly "to remove PCR duplicates". It's also not true that "Properly recovery of loci is therefore not possible with read stacking approaches."

I installed using the conda package and `run_discoSnpRad.sh` did not work out-of-the-box. I had to edit clustering_scripts/discoRAD_clustering.sh and change #!/bin/sh to #!/bin/bash, otherwise `run_discoSnpRad.sh -r fof.txt -S` would crash. This is on a stock Ubuntu linux machine, nothing fancy.

It's good practice to ship working example data with the code to make it easier for people to try out. I tried running the test data (https://github.com/GATB/DiscoSnp/tree/master/test) with discoSnpRad defaults and it returned 3 snps in the vcf. A 3 locus dataset with 1 snp per locus isn't close enough to biological reality to give people a good sense of what's going on. I generated my own simulated dataset (which assembled fine in both stacks and ipyrad), very clean 1000 loci with a reasonable level of variation (nucleotide diversity ~0.002), and discoSnpRAD returned 0 snps in the vcf. Update: I figured out what's going on, the fq.gz files need to be demultiplexed to samples. I think it's important to mention this in the manuscript and in the documentation because it led me to an extreme amount of confusion for a while.

Once it started working it finished fine and claimed to place the results in a file which doesn't exist:
* * *
Short read connector finished
results in:
discoRad_k_31_c_3_D_0_P_5_m_5_raw_filtered_simpler.txt
Contact: pierre.peterlongo@inria.fr
* * *
Reviewer 2 ·

Basic reporting

no comment

Experimental design

no comment

Validity of the findings

no comment

Additional comments

In their manuscript ‘DiscoSnp-RAD: de novo detection of small variants for population genomics’, Gauthier and coauthors present a novel method for RADseq data analysis. The method adapts the DiscoSNP++ (Peterlongo et al. 2017) reference-free SNP discovery and genotyping approach to the specificities of RADseq data, thus allowing its application to a data type that is widely used, and often so in systems where no reference genome is available. They compare the performance of the method against two methods commonly used for RADseq analysis, finding that their method performs similarly to existing ones in terms of recall and precision of SNPs and very favorably in terms of computational requirements. They then use it to re-examine a real insect dataset (Chiastocheta flies; Ronikier & Alvarez 2017).

The presented work is relevant and of interest to the community, the analyses appear to be well executed, and the manuscript is generally clear and well written.

I nevertheless have comments on specific elements of the manuscript, listed below:

1. The authors state on multiple occasion (L28, L87, L293, L367, etc.) that their method successfully bypasses the need for any similarity parameter. Since this is a core aspect of the method, I believe the authors should develop their current discussion of this property (L369 ‘[parameter choices] impact the number and type of detected variants’), that they should further explain how this property arises eg. in the introduction, and that they should discuss the drawbacks of the approach.

2. Figure 3: it would be very desirable to add a panel showing how k-mer size influences some general statistic(s) for the real dataset, in addition to the results given for the simulation dataset.

3. It is repeatedly affirmed that clustering in Stacks and ipyrad require ‘all-versus-all read alignments’ and thus that the computational requirements ‘increase quadratically’. This is wrong; both methods use k-mer-based heuristics. Please review the relevant literature and update the text in all relevant places.

4. The most recent Stacks and ipyrad publications should be cited (respectively 2019 and 2020). Also, for Uricaru et al. the final citation appears to be 2015.

5. For simulations, the authors should justify why they compared the results of the different methods to ‘'the standard protocol of DiscoSnp++ when a reference genome is provided’ (L230) rather than to the actual simulated genotypes, especially as this has the potential to bias the results in favor of DiscoSnp-RAD.

6. L262 Authors should state that they have explored a relevant range of values of K and that K=7 was the most biologically meaningful (cf. Novembre, 2016).

7. L276 Variant calling fundamentally has two components: the discovery of polymorphisms, and their genotyping in each individual. The authors should justify why they report recall and precision for SNPs but not for individual genotypes (or give the figures for genotypes).

8. L332 ‘The STRUCTURE analysis of this dataset tended to identify two populations and assigned 31 samples to one of them and 9 to the other (Fig.4)’ I disagree with this statement; all samples are primarily associated with the same component. The signal the authors refer to may exist, but STRUCTURE does not seem to be the right approach to describe it. The authors may want to perform a PCA and consider the PC(s) most correlated with longitude and update figure 4.

9. Authors should make their custom scripts available, especially those related to benchmarking and simulations.

Minor comments:

10. L20 ‘DiscoSnp-RAD […] avoids this pitfall since variants are detected by exploring the De Bruijn Graph built from all the read datasets.’ Phrasing could be improved for non-specialist readers

11. L47 ‘The sequencing output is thus composed of hundreds of thousands of reads’ Millions?

12. L52 ‘mainly’ is unwarranted

13. L79 ‘We propose an adaptation of the DiscoSnp++ approach to [RAD-Seq data]. After validation tests […]’ Authors could insert a short description of how their method differs from the previously described DiscoSNP++ implementations (cf. Methods).

14. L88 Please proofread

15. L91-93: Please proofread

16. L130 Please clarify whether ‘variants’ comprises SNPs, indels, or both.

17. L170 The default minimum per-allele depth threshold of 3 seems quite high; this may create bias against heterozygous genotypes. This is also redundant with the binomial likelihoods approach, which is more relevantly combined with a filter on total coverage.

18. L192-200 The definition of edges (and thus of loci) could be clarified, especially as this functionality is going to be of particular interest to users.

19. L220 ‘SNPs were randomly generated at a rate of 0.01%’ Please confirm that this is not 0.01/1%

20. L235 Authors should state why they did not compare against dDocent (Puritz2014).

21. L240 Please proofread

22. L296 “between samples” is confusing and should be removed; samples are typically diploid.

23. L310 Which of the filtering criteria is the primary cause of the removal of 95% of SNPs?

24. L329 ‘The same run of DiscoSnp-RAD obtained with the 259 samples was used, only the vcf was limited to the 40 sample columns of this species and the same filters were applied on this reduced dataset’ Please improve this sentence.

25. L383 Please clarify whether Disco-SNP itself is able to remove PCR duplicates, as for this purpose alignments to a reference are usually needed.

26. L391 Please clarify why DiscoSNP-RAD is more relevant than DiscoSNP++ for sequence capture datasets

·

Basic reporting

no comment

Experimental design

Line 253. Despite both the main text and the documentation provided in the github repository may suffice to the average experienced bioinformatician, I think maybe some beginners could find it a little hard to replicate the results when they reach the filtering part, after the execution of DiscoSnp-RAD itself. That’s because even the scripts to do so are available on github, there aren’t any instructions on which and how or when to use each one of them, besides the texts inside the scripts themselves. I understand that they aren’t part of DiscoSnp-RAD itself, and even that one could use any other tools provided around the internet to perform their tasks, but I think your pipeline could have a greater reach if even a brief explanation around the scripts was given either in the paper or in the software documentation.

Validity of the findings

Line 327. It’s great to see that DiscoSnp-RAD is capable of detecting such a shallow, intraspecific signal, proving the other positive results aren’t just coming from greater differences between lineages. As far as I could find on the original paper by Suchan et al 2017, it seems to be an unpublished result regarding C. lophota, and if it doesn’t it has to be pointed out more clearly. But considering it is indeed a new result, it wasn’t clear to me whether the intention was suggest DiscoSnp-RAD can be more sensitive than PyRAD (the software used on that work) or if you ran STRUCTURE only with the SNPs got from DiscoSnp-RAD, unlike what was done in the previous section in the paper (line 309).

Additional comments

I was able to successfully replicate something similar to your results on my own data, when comparing results between DiscoSNP-RAD and Pyrad. I’d like to say it was great, and even in a quite small dataset the processing time with DiscoSnp-RAD was twice as fast as Pyrad’s. Moreover, I could take a close look on what the users will find when they start using this great new tool, and I have a couple suggestions.

- I think you should consider incorporating short_read_connector (or an equivalent) into DiscoSnp-RAD. It’s was not very difficult for me to make everything work, but I have the impression that it could discourage more inexperienced users the way it is now, due to issues with dependencies;
- At first I couldn’t use the clustering option (-S flag), and just after looking into the code I realized it HAS to come first in arguments order, when running the DiscoSnp-RAD. I didn’t find any information regarding that in the documentation. Either way, isn’t hard to make the position of arguments flexible, what certainly could save users some trouble.
- I think you could give DiscoSnp-RAD its own exclusive user’s manual and expand it a little, including not only the equivalent to what you have in DiscoSnp++’s part but also the information I talked about in other topic. Many researchers – myself included – will make use of the RAD part only.

---

## Round 0.2 · Minor Revisions

Dear authors,

Thank you for your revision. You did quite a thorough job. Both reviewers and I are happy, however, reviewer 1 takes issues with your claim that DiscoSNP-RAD does not need any parameter tuning for optimal SNP extraction. Given that RAD data are quite heterogeneous among organisms and/or clades, and both Stacks and iPyRAD have tuning parameters permitting “optimal” extraction of information, it is hard to accept at face value that no parameter tuning is necessary in DiscoSNP-RAD. Ideally you should prove this claim by varying within a reasonable range your internal parameter and then show you get the same results on your simulated dataset, and that datasets simulated under different scenarios will result in an optimal SNP output when default parameters are used as when the parameter are varied. If this is beyond what you want to do with this paper, I suggest you tone down this claim.

Personally, I think it would be very useful if there were an option for DiscoSNP-RAD to also output haplotype information, and not just SNP in VCF. But perhaps this could be a future development.

Otherwise, I am happy with your MS, and I look forward to your response and revision.

Tomas Hrbek

Reviewer 1 ·

Basic reporting

L194: It's not clear what 'binary variables' are being referred to.

Experimental design

Are there options for producing haplotypes rather than just snps (i.e. recovering the full RAD locus information)? It seems not, but this should be clarified. Some downstream tools require haplotype data.

I would recommend including in the manuscript information about how R1 and R2 are handled for paired-end data. I understand that if R1 and R2 are overlapping the linkage information will be retained, but am somewhat fuzzy on how many bp they must overlap by for this to be true. Also, if they are not overlapping are they treated as independent? In response to reviewers it is stated that they "create independent connected components in the de Bruijn graph", but it's not clear to me if this means the linkage information is retained. In particular, "independent connected" does not clarify this for me.

I thank the authors for clarifying the Phi coefficient and 'rank' in the manuscript, and I appreciate the information given in response to Reviewer 1s question about low rank variants, which I find interesting, but which I do not find in the revised manuscript. In particular, I would recommend discussing why "'heterozygous in all' variants should be very rare", and what might be the downstream ramifications of removing these variants.

The text introduced to address reviewer 1s comment on Line 136 is still ambiguous. If you "constrain the last 3-mer of both paths to be identical" it is still not clear how this "does not prevent to detect loci containing such variants." Instead of "we also constrain the last 3-mer of both paths to be identical" I think it would be more accurate to say "the last 3-mer of both paths are excluded from the analysis.", or something like this.

Validity of the findings

RADSeq datasets are notorious for their heterogeneity along multiple axes: missingness, read depth per locus within samples, sample depth per locus, not to mention nucleotide diversity and other aspects of the genomic architecture of the focal species, and the peculiarities of any given library prep protocol, etc, etc. There is a reason other assembly tools have parameters and this reason is because many RADSeq datasets are peculiar. In truth of fact, Figure 3 shows that changes in parameter values actually do impact the results for DiscoSnp-RAD, contrary to what the authors insist on. The asymptotic behavior of precision and recall for each parameter shown in Figure 3 are relevant only to THIS ONE simulated dataset that is tested. Heterogeneity in real datasets will require tuning DiscoSnp-RAD parameters in much the same way as other RADSeq assemblers, in order to optimize results. I think a lot more work would need to be done to demonstrate this is not the case, for example analysis of simulated data generated under a wider array of conditions. It would also be nice to see something like figure 3 for the empirical data, varying parameters and plotting number of SNPs/loci recovered. This would go a long way toward strengthening their claims, if the recovered SNPs do not change with varying parameters for the real data. My previous recommendation was to emphasize that "rather than not requiring fine tuning, DiscoSnp-RAD tries to choose sensible defaults," and I stand behind this recommendation, given the current state of the manuscript.

For the comparison to other tools it occurs to me that the 1% sequencing error in the simulations is more than just unrealistic. It is also probably what's causing the significant difference in performance measured in STACKS and ipyrad. This insight is supported by the results described in the "Breakthrough in running time" section which shows that on real data STACKS and ipyrad perform significantly more similarly to DiscoSnp-RAD than with the simulated data. Rather than redo the simulations, I recommend adding something to the discussion about the choice of sequencing error rate for the simulated data and how this might impact performance of the various assemblers.

The manuscript still states: "The phylogeny realized with RAxML on the 4,364 SNPs obtained after filtering, is congruent with the one obtained by Suchan and colleagues [25] (Fig.4).", which the authors agree is actually false. This sentence literally says that the RAxML trees are congruent between studies. If this is not the case this wording should be adjusted accordingly.

L407: Proper handling of GBS data involves accounting for the fact that reads from any given locus can come from either the forward or the reverse strands. It's not clear this method considers strandedness when building the DBG. I am not a DBG expert, so perhaps reverse complement matching consideration is made during this process, and if so I apologize. If revcomp is not considered then I would remove the reference to GBS here.

Additional comments

During the course of my second review I find that the authors satisfactorily addressed many of my comments from the first review. I also must emphasize that DiscoSnp-RAD actually is quite fast, and in independent experiments I never found a case where STACKS or ipyrad were faster, this is totally true. However, I have several comments of secondary concern enumerated below, and my primary concern remains the insistence on "Robustness with respect to parameters", which I elaborate on in the "Validity of findings" section below.

Reviewer 2 ·

Basic reporting

no comment

Experimental design

no comment

Validity of the findings

no comment

Additional comments

The authors have honestly and thoroughly addressed the concerns raised by me and the other reviewers. In particular, the expansion of figure 3 and the work on the Methods section are welcome additions. I believe that they present a strong manuscript and that their method DiscoSNP-RAD will be very valuable to the community.

---

## Round 0.3 · accepted · Accept

Dear authors,

Thank you for your revision, and for addressing the first reviewer’s comments. I am happy with your revision, and I am happy to recommend accepting your MS for publication.

Congratulations on an excellent job, I am sure that DiscoSNP-RAD will turn out to be a widely used tool.

Tomas Hrbek